# Bi- and uniciliated ependymal cells define continuous floor-plate-derived tanycytic territories

Zaman Mirzadeh[1], Yael Kusne[1], Maria Duran-Moreno[2], Elaine Cabrales[1], Sara Gil-Perotin[2], Christian Ortiz[3], Bin Chen[3], Jose Manuel Garcia-Verdugo[2], Nader Sanai[1] & Arturo Alvarez-Buylla[4]

Multiciliated ependymal (E1) cells line the brain ventricles and are essential for brain homeostasis. We previously identified in the lateral ventricles a rare ependymal subpopulation (E2) with only two cilia and unique basal bodies. Here we show that E2 cells form a distinct biciliated epithelium extending along the ventral third into the fourth ventricle. In the third ventricle floor, apical profiles with only primary cilia define an additional uniciliated (E3) epithelium. E2 and E3 cells' ultrastructure, marker expression and basal processes indicate that they correspond to subtypes of tanycytes. Using sonic hedgehog lineage tracing, we show that the third and fourth ventricle E2 and E3 epithelia originate from the anterior floor plate. E2 and E3 cells complete their differentiation 2–3 weeks after birth, suggesting a link to postnatal maturation. These data reveal discrete bands of E2 and E3 cells that may relay information from the CSF to underlying neural circuits along the ventral midline.

[1] Division of Neurological Surgery, Barrow Neurological Institute, Phoenix, Arizona 85013, USA. [2] Laboratory of Comparative Neurobiology, Instituto Cavanilles, CIBERNED, Universidad de Valencia, Valencia 46980, Spain. [3] Department of Molecular, Cell and Developmental Biology, University of California, Santa Cruz, California 95064, USA. [4] Department of Neurological Surgery and The Eli and Edythe Broad Center for Regeneration Medicine and Stem Cell Research, University of California, 35 Medical Center Way, Room RMB-1036, Campus Box 0525, San Francisco, California 94143, USA. Correspondence and requests for materials should be addressed to Z.M. (email: zaman.mirzadeh@barrowbrainandspine.com) or to A.A.-B. (email: AlvarezBuyllaA@ucsf.edu).

The ependymal epithelium lines the adult brain ventricles, where it plays key roles in cerebrospinal fluid (CSF) flow and brain homeostasis[1,2]. Cilia, a distinguishing feature of ependymal cells, are critical to their development and function[3–5]. Using high-resolution imaging of the ependymal surface[6], we previously identified an ependymal cell apical profile in the lateral ventricles (LVs), with only 2 (9 + 2) cilia. These biciliated ependymal cells, called E2 cells, have two large basal bodies with elaborate raceme-like appendages. The function and origin of E2 cells remains unknown. These cells are extremely rare in the LVs, making them difficult to study.

Ependymal cells' planar orientation[3,4] is essential for propelling CSF[7] and establishing chemorepellent gradients guiding migratory neuroblasts in the adult brain[8]. Ependymal cells are integral to the pinwheel organization and function of the adult germinal niche in the ventricular-subventricular zone[6,9]. Tanycytes, a subpopulation of ependymal cells bearing long basal processes, are abundant in the third ventricle and line circumventricular organs[10], where fenestrated capillaries permit neuroendocrine cross-talk[11,12]. For example, tanycytes of the median eminence have critical functions in energy balance that, when disrupted, result in obesity[13–16]. However, the organizing principles and developmental patterning that establish this ependymal heterogeneity are unknown.

Ependymal cells are derived from radial glia[17], the embryonic neural stem cells[18]. Recently, these stem cells were shown to have remarkable heterogeneity[19], with regionally restricted potential to produce various neuronal subtypes. As descendants of radial glia, ependymal cells may inherit this regional identity, which then determines their heterogeneity. Although ependymal heterogeneity, including two types of tanycytes (α and β), has been documented in many species[20–24], the embryonic origin of this heterogeneity has not been studied.

Here we identified a distinct epithelium of biciliated (E2) ependymal cells that extended along the ventral third ventricle (3 V), cerebral aqueduct (CAq) and fourth ventricle (4 V). Ultrastructural and molecular marker characterization identified E2 cells in the 3 V as α-tanycytes. In the floor of the 3 V, we found a third apical profile with a (9 + 0) primary cilium characterizing another ependymal cell type (E3), which corresponded to β-tanycytes. These observations link functional subtypes of tanycytes to defining apical characteristics of E2 and E3 cells. Furthermore, we show that E2 cells extend as a continuous epithelium along the floor of the CAq and 4 V. We provide molecular markers that distinguish these different epithelia and lineage-traced E2 and E3 cells to embryonic progenitors expressing sonic hedgehog (Shh), suggesting they are floor-plate derivatives. The work demonstrates that apical profile heterogeneity among ependymal cells may be traced to an essential tissue-organizing centre in the embryo and shifts our perspective of the ependyma from a simple ventricular lining to an organized vestige of development with implications for its diverse functions.

## Results

**Apical profiles define ependymal territories.** E2 cells comprised <5% of cells contacting the LV[6]. We investigated whether E2 cells were more common in other ventricles by mapping their location in the walls of the third ventricle (3 V; Fig. 1a) and the floor of the fourth ventricle (4 V) (Fig. 1h). Whole mounts from these walls were immunostained with γ-tubulin and β-catenin antibodies, the ependymal surface was imaged sequentially to cover the entire surface and the location of E cell types were mapped.

In contrast to sparse E2 cells found in the LVs, a striking E2 distribution was observed in the 3 and 4 V (Fig. 1a,h).

Dorsolaterally, E1 cells with sparsely intermixed E2 cells covered these walls, but there was a sharp margin approaching the ventral midline where the epithelium changed abruptly (Fig. 1b,i). Beyond this margin in both the 3 and 4 V, the epithelium was composed almost uniformly of E2 cells, with little intermingling of E1 cells. This E2 epithelium was distinguished not only by unique basal bodies, but by highly interdigitating apical membranes marked by β-catenin (Fig. 1c,i). To examine E2 cilia, we stained wholemounts for γ-tubulin, to identify basal bodies, and acetylated tubulin, to label cilia. E2 cells in the 3 and 4 V differed in their cilia complement (Fig. 1e,f,j,k): in the 3 V, 37% were biciliated, 47% were uniciliated and 16% had no cilia (104 cells from 3 mice); in the 4 V, 80% were biciliated, 18% were uniciliated and only 2% had no cilia (207 cells from 3 mice). Cilia length among E2 cells in the 3 V versus 4 V was similar (mean ± s.d.): 3 V (12.3 ± 1.8 μm) versus 4 V (12.0 ± 1.9 μm; $P = 0.6$, two-tailed unpaired $t$-test). Apical surface size, however, differed (mean ± s.d.): 3 V ($74 ± 26\ \mu m^2$) versus 4 V ($84 ± 21\ \mu m^2$; $P = 0.01$, two-tailed unpaired $t$-test).

Based on apical organization, a third epithelial cell type was observed, which we called E3 cells. E3 cells had a simple basal body, a smaller apical surface area than E2 cells ($25 ± 6\ \mu m^2$) and more intense β-catenin expression at their intercellular junctions (Fig. 1d). These cells almost always (95% of 521 E3 cells from 3 mice) extended a single, short cilium (5.0 ± 1.0 μm) from their apical surface (Fig. 1e,g). The relationship among E1, E2 and E3 epithelia in the 3 V is shown in Fig. 1a: moving ventrally along the 3 V wall, the transition from E1 to E2 cells was followed by another transition, from E2 to E3 cells (Fig. 1d). Occasionally in the E3 territory, E2 cells and, more rarely, E1 cells were observed. Two regions in the ventral 3 V contained E3 cells: along the preoptic recess (Fig. 1a, preoptic area (POA)) and along the infundibular recess, near the ventromedial hypothalamus and median eminence (Fig. 1a). Caudally, the E2 epithelium covering the ventromedial hypothalamus continued uninterrupted along the ventral midline, up the posterior limb of the 3 V through the CAq and into the 4 V floor. Performing whole mounts of the CAq was difficult due to its small diameter. We confirmed the continuity of the E2 epithelium from 3 to 4 V in γ-tubulin-stained serial coronal sections (Supplementary Fig. 1). In addition, the 4 V E2 epithelium was continuous with the E2 epithelium encircling the spinal cord central canal[25]. Identifying this distinct epithelium from 3 to 4 V facilitated ultrastructural and molecular marker characterization of these cell types.

**E2 and E3 cells have distinct basal bodies and cilia.** By transmission electron microscopy, E3 cilia corresponded to primary cilia with (9 + 0) microtubule structure (Fig. 2a,d). The cilium was nucleated by a basal body that had an associated, orthogonally oriented centriole. E2 cilia in the 3 V (Fig. 2b,e) and 4 V (Fig. 2c,f) had a (9 + 2) microtubule structure, corresponding to motile cilia, as in E2 cells of the LVs and E1 cells[6]. However, unlike E1 basal bodies, which were small and had simpler organization, E2 basal bodies in the 3 V (Fig. 2g–p) and 4 V (Fig. 2q–z) elaborated raceme-like networks of dense particles from their sidewall. Associated with E2 basal bodies were 18 nm cytoskeletal filaments (Fig. 2j,n,y, black arrowhead) and striated ciliary rootlets (Fig. 2j,r, white arrowhead). E2 cells had extensive apical membrane interdigitations with cell–cell junction complexes, including adherens and gap junctions. E2 cells in the 4 V displayed cytoskeletal anchorage of many mitochondria to the proximal end of their basal bodies (Fig. 2x, black arrow), also found in E1 cells. This analysis showed that E2 and E3 cells differed in their apical ciliary machinery, but E2 cells' apical organization was largely similar throughout the ventricular system.

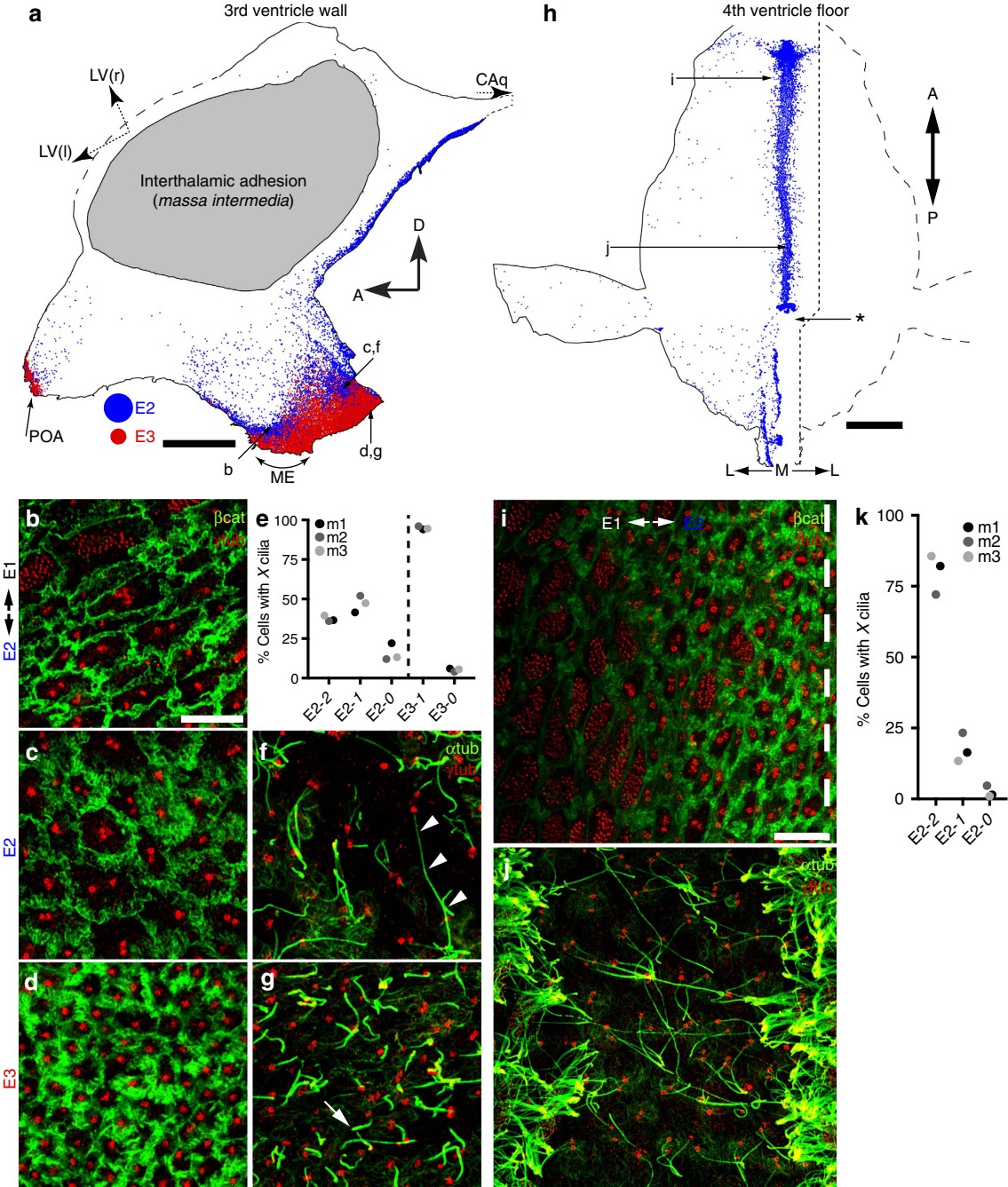

**Figure 1 | Territories of E2 and E3 cells in the third and fourth ventricles. (a,h)** Whole mount maps of the 3 V wall (**a**) and 4 V floor (**h**) were derived from tiled confocal images that reconstructed the ependymal surface, stained with γ-tubulin and β-catenin antibodies, from a single mouse. Blue dots indicate E2 cells and red dots indicate E3 cells. (**a**) Dashed arrows through the foramen of Munro point towards right (r) and left (l) lateral ventricles (LV), and a third dashed arrow points to the CAq; compass shows anterior (A) and dorsal (D) directions; letters indicate position of corresponding panels. ME, median eminence; POA, preoptic area. Scale bar, 1 mm. (**b–d**) Confocal images of the 3 V surface stained with γ-tubulin (red) and β-catenin (green) antibodies showing the border between E1 and E2 cells (**b**), the E2 epithelium (**c**) and the E3 epithelium (**d**). Scale bar, 10 μm (**b–d,f,g**). (**e**) Percentage of E2 cells with 2, 1 or 0 cilia and E3 cells with 1 or 0 cilia in the 3 V wall. One hundred and four E2 cells and 521 E3 cells analysed from $N = 3$ mice (m1–m3). (**f,g**) Confocal images of the 3 V surface stained with γ-tubulin (red) and acetylated α-tubulin (green) antibodies showing long cilia of biciliated or uniciliated E2 cells (**f**, arrowheads) and short primary cilia on E3 cells (**g**, arrow). (**h**) One side of the 4 V floor was mapped completely (left) and the contralateral side (right) to ~1 mm beyond the midline (indicated by dashed line). Compass shows anterior (A), posterior (P), midline (M) and lateral (L) directions; letters indicate position of corresponding panels; asterisk marks a position in the floor posterior to which the E2 distribution changed from a thick concentrated midline band to thinner strips of E2 cells not limited to the midline. Scale bar, 1 mm. (**i**) Confocal image of the 4 V floor surface stained with antibodies to γ-tubulin (red) and β-catenin (green) showing the border between E1 and E2 cells just lateral to the midline. Dashed line indicates the midline. Scale bar, 10 μm (**i,j**). (**j**) Confocal image centred on the 4 V floor midline stained with antibodies to γ-tubulin (red) and acetylated α-tubulin (green), revealing the midline corridor of mostly biciliated E2 cells surrounded by multiciliated E1 cells. (**k**) Percentage of E2 cells with 2, 1 or 0 cilia in the 4 V floor. 207 E2 cells analysed from $N = 3$ mice (m1–m3).

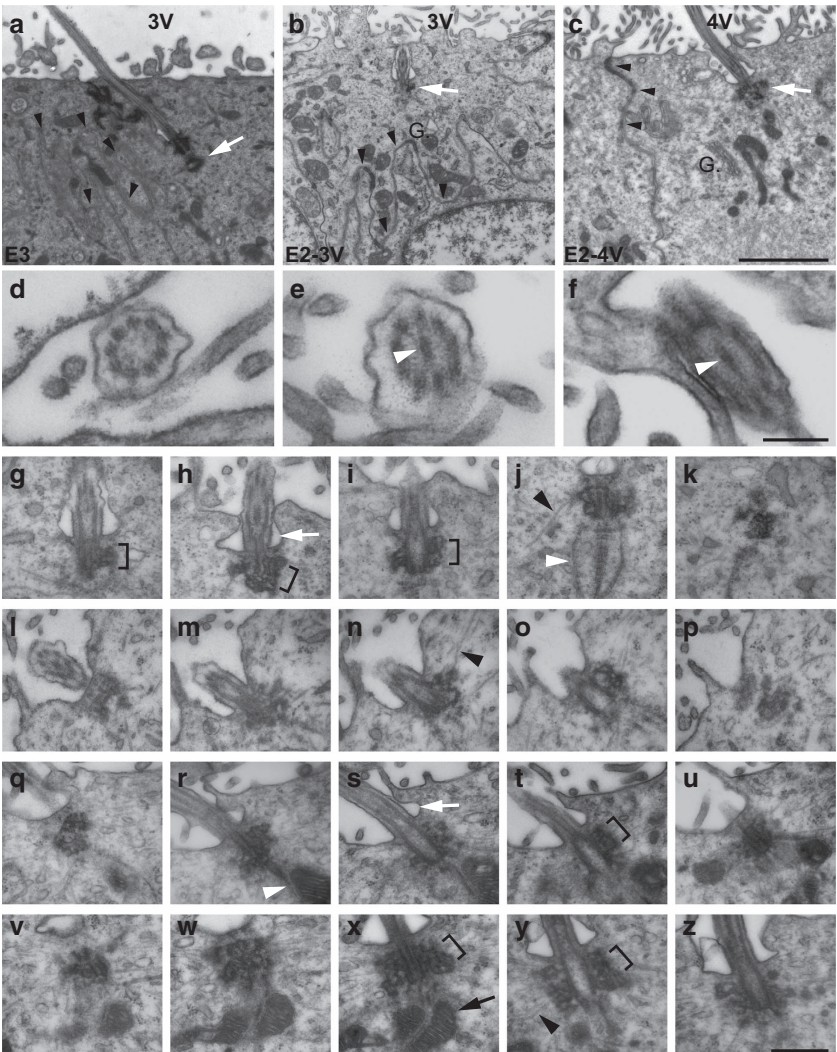

**Figure 2 | Unique cilia and basal body organization of E2 and E3 cells in the 3 V and 4 V.** (**a**) EM micrograph of an E3 cell apical compartment in the 3 V with the typical centriole orthogonal to the basal body (white arrow). Arrowheads indicate interdigitating membranes. (**b,c**) EM micrographs of E2 cells in the 3 V (**b**) and 4 V (**c**) with large dense basal bodies (white arrows). G., Golgi apparatus. Scale bar, 2 mm (**a–c**). (**d–f**) Higher magnifications of cross-sections of cilia in E3 (**d**), 3 V E2 (**e**) and 4 V E2 (**f**) axoneme revealing the microtubule structure: E3 cilia are 9 + 0 (**d**), E2 cilia are 9 + 2 (3 V E2 in **e** and 4 V E2 in **f**; white arrowheads indicate central pair of microtubules). Scale bar, 200 nm (**d–f**). (**g–p**) Serial ultrastructural reconstruction of cilia and associated basal bodies of an E2 cell in the 3 V. Extensive, dense raceme-like appendages (brackets) emanate laterally from the basal bodies. Black arrowheads indicate 18 nm-thick filaments associated with the basal body and white arrowheads indicate long ciliary rootlets. White arrow indicates the ciliary pocket at the base of E2 cilia. (**q–z**) Serial ultrastructural reconstruction of cilia and associated basal bodies of an E2 cell in the 4 V. E2 cells in the 4 V show features in common with 3 V E2 cells: raceme-like appendages (brackets), 18 nm filaments (black arrowhead), ciliary rootlets (white arrowhead) and ciliary pocket (white arrow). In contrast to the 3 V, E2 cells in the 4 V displayed cytoskeletal anchorage of mitochondria to their basal body (**x**, black arrow). Scale bar, 500 nm (**g–z**).

**Molecular markers distinguish E cell types**. We immunostained 3 and 4 V whole mounts with antibodies to several molecular markers, to characterize differences in E cell types. We found ubiquitous expression of the intermediate filament proteins vimentin and nestin by E1, E2 and E3 cells, seen in low-magnification images of the posterior ventral 3 V (Fig. 3a,b,k) and ventral midline 4 V (Fig. 3f,g,l). High-magnification images of whole mounts co-labelled with γ-tubulin, to identify the E cell type, and the respective molecular marker revealed differences in apical intermediate filament organization (Fig. 3a,b,f,g, insets labelled II correspond to E2 region and insets labelled III correspond to E3 region). S100β, a calcium-binding protein, and CD24, a cell surface sialoglycoprotein, showed a different pattern: these were expressed throughout the E1 and E2 epithelia, but were notably absent in the E3

epithelium, with the E2–E3 margin sharply demarcated in low-magnification images of the 3 V (Fig. 3c,d,h,i). High-magnification images of co-staining with γ-tubulin (for S100β) or acetylated-tubulin (for CD24) confirmed E1 and E2, but not E3, labelling (Fig. 3c,d,h,i insets). S100β was localized to the cytoplasm, whereas CD24 was expressed on the apical surface and cilia. Glial fibrillary acidic protein (GFAP), another intermediate filament found in some astrocytes and some E1 cells, was expressed along a stripe on the ventral 3 V surface (Fig. 3e) and 4 V floor (Fig. 3j). The vast majority of GFAP + ependymal cells in the ventral 3 V were E2 cells and the GFAP + stripe corresponded to the E2 epithelium, with only a fraction of E1 cells (16%) and almost no E3 cells expressing GFAP (Fig. 3e insets). In the 4 V, the midline GFAP + stripe also corresponded to the E2 epithelium (Fig. 3j inset).

Whole mount expression patterns were matched to corresponding regions in immunostained serial coronal sections (Supplementary Figs 2 and 3), to characterize the basal morphology of ependymal cell types. E2 and E3 cells had long basal processes, labelled by vimentin, Nestin and GFAP (E2 only). E3 processes primarily penetrated the median eminence and arcuate nucleus, with many terminating at the pia. E2 processes penetrated diencephalic and brainstem regions based on their location: rostrally, they extended into the dorsal arcuate and ventromedial nuclei; caudally within the 3 V, they penetrated the caudal dorsomedial nucleus; along the posterior limb of the 3 V and CAq, they extended into the periaqueductal gray and posterior hypothalamic nucleus; and at the level of the 4 V, these processes were found in caudal dorsal raphe nuclei and penetrating the gigantocellular reticular nucleus. In sections and whole mounts, these cells could be distinguished by GFAP/CD24/S100β expression in E2 cells and absence in E3 cells.

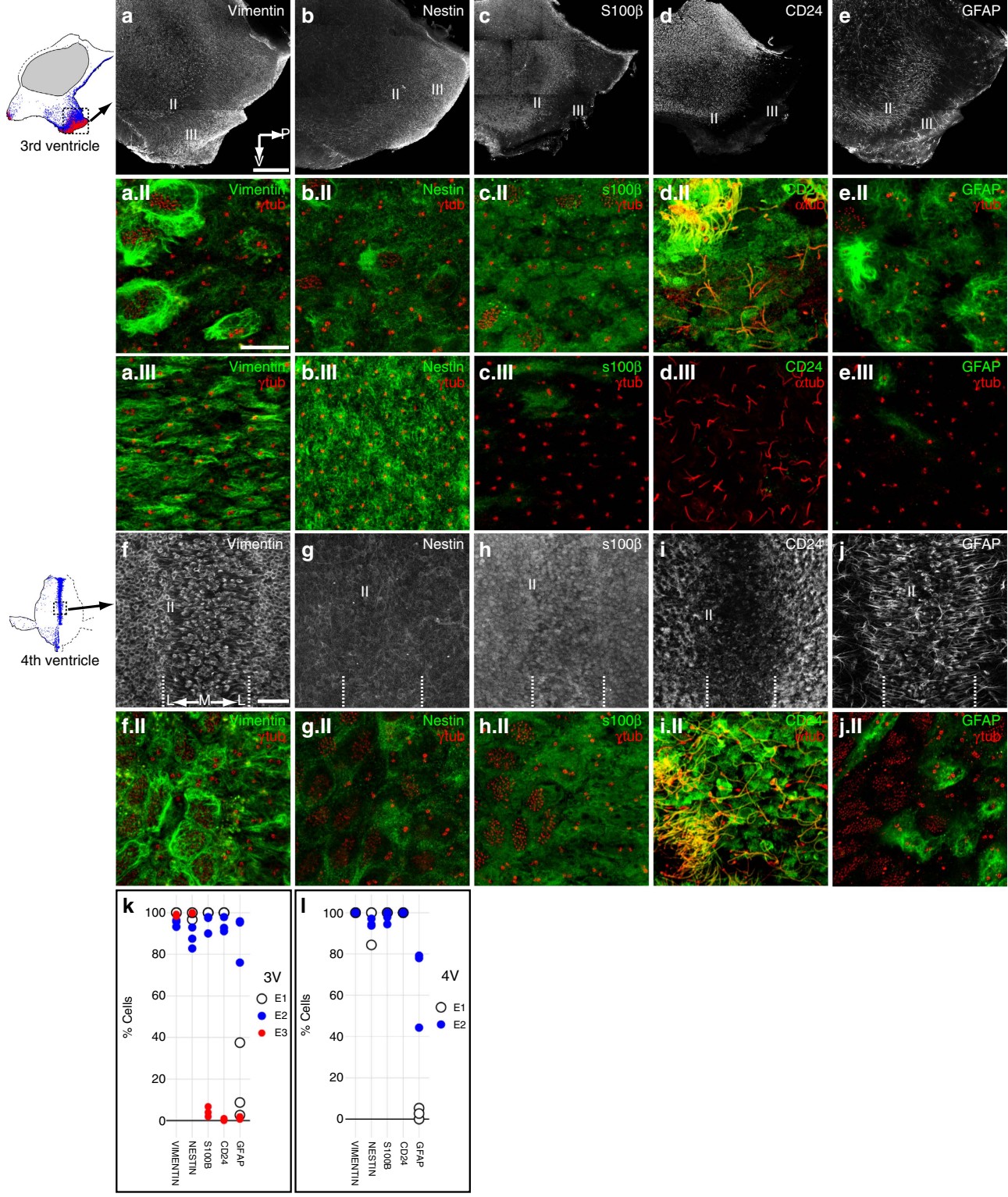

**Fezf2 is expressed by E3 and a subpopulation of E2 cells.** To find a marker that better identified E3 cells, we searched the GENSAT database[26] and the Allen Brain Atlas[27] for genes differentially expressed in the ventral third ventricle. We found several candidates, including Delta-like 1 (Dlk1), solute carrier family 16 member 2 (Slc16a2), and collagen type 23α1 (Col23a1), but took interest in forebrain embryonic zinc finger 2 (Fezf2). Fezf2 is a transcription factor known for its role in specification of subcerebral projection neurons, but it is also expressed in the 3 V lining and has been shown to be dynamically regulated during mouse hypothalamic development[28]. Whole mounts from Fezf2-GFP bacterial artificial chromosome (BAC) transgenic mice[26], which express green fluorescent protein (GFP) under control of the Fezf2 promoter, showed GFP expression in the E3 region (Fig. 4a). These cells had typical E3 morphology, including a basal process terminating at the pia (Fig. 4b). Co-labelling with γ-tubulin and β-catenin, to identify E cell types, and GFP revealed that nearly all E3 cells (99.6% of 760 cells from 3 mice, >200 cells per mouse) expressed Fezf2. In addition, a subset of E2 cells in the 3 V (40.2% of 307 cells from 3 mice, >100 cells per mouse) expressed Fezf2 (Fig. 4c–e). In combination with the markers above, we could now characterize E1 cells as primarily CD24 + GFAP-Fezf2 − , E2 cells as CD24 + GFAP + Fezf2 + / − and E3 cells as CD24-GFAP-Fezf2 + (Fig. 4f,g). Examination of these expression patterns at 2 months and 10 months of age revealed that the territories occupied by these epithelial cells were relatively stable in size and location (Fig. 4h,i). Finally, we analysed a different transgenic mouse line, expressing CreER under control of a 2.7 kb region of the Fezf2 promoter[29], to corroborate Fezf2-driven GFP expression in the BAC transgenic line, as well as Fezf2 *in situ* hybridization data available from the Allen Brain Atlas[27]. Fezf2-CreER mice were crossed with RCeGFP mice (Cre reporter line expressing enhanced GFP); offspring were administered tamoxifen at P2 and killed at P90. GFP-labelled cells were observed in whole mounts in the ventral 3 V within the E3 territory (Supplementary Fig. 4a). These cells exhibited both long basal processes reaching the pia (Supplementary Fig. 4b) and small apical surfaces with a single basal body and primary cilium (Supplementary Fig. 4c,d), corresponding to E3 cells. Furthermore, their relatively restricted position within the ventral 3 V suggested that this cohort of cells, which were labelled 3 months earlier, had not translocated into more dorsal positions within the 3 V epithelium. This understanding of molecular marker expression patterns and the relatively stable position of cells within the epithelium facilitated our developmental analysis.

**Postnatal differentiation of E2/E3 cells.** E1 cells are derived from radial glia[17] and differentiate postnatally from these cells, bearing a single primary cilium, into a cell with multiple motile cilia[30]. To determine the time course of E2 and E3 cell differentiation, 3 and 4 V whole mounts from embryonic day 14.5 (E14.5) to postnatal day 35 (P35) were stained with γ-tubulin and β-catenin antibodies (Fig. 5 and Supplementary Fig. 5). At birth (P0), apical profiles with a single basal body and smooth membranes, consistent with radial glia[6], uniformly covered the ventral 3 V surface. Interestingly, cells in the putative E2 territory were already expanding their apical surface. Similar uniciliated apical surfaces were observed along the 4 V ventral midline, whereas laterally multiciliated E1 cells had begun differentiating (Fig. 5a–c). At P7, we observed a subset of cells in the 3 V E2 territory with two basal bodies, as well as differentiating E1 cells anterior to this region. In the 4 V E2 territory, some apical surfaces also started to exhibit two basal bodies (Fig. 5d–f, arrows). By P15, apical membranes in the 3 V E3 region had developed interdigitations and most cells in the 3 and 4 V E2 regions had two basal bodies (Fig. 5g–i). By P35, the ependymal surfaces of both ventricles had matured to their adult form, with further interdigitations and the full basal body complement throughout the E3 and E2 epithelia (Fig. 5j–l). A parallel developmental series of whole mounts, stained with γ-tubulin and acetylated tubulin, showed that cilia in these cell types matured with a similar time course (Supplementary Fig. 5). At P0, single cilia in the 3 V putative E2 region ((mean ± s.d.), 4.7 ± 1.5 μm) were longer than those in the E3 region (2.0 ± 1.0 μm), suggesting that E2 cells had initiated their programme to build longer cilia, before the appearance of two cilia. In the 4 V, single cilia (6.3 ± 2.1 μm) along the midline were longer than expected for radial glia (∼3 μm, ref. 6). By P7 the cilia in each region had grown (E3 − 3 V: 5.0 ± 1.0; E2 − 3 V: 6.6 ± 1.6; E2 − 4 V: 9.5 ± 2.2 μm) and by P15 biciliated E2 cells were seen, with mature cilia length and number reached by P35 (3 and 4 V whole mounts studied from 2 mice per age; >30 cells per region per mouse).

Next we determined how apical surface maturation corresponded with the differentiation of radial glia into E2 and E3 cells. In whole mounts, we examined the postnatal expression of RC2, a marker for radial glia[31], and GFAP, to identify E2 cells. Over the first 10 days postnatally, 3 V (Fig. 6a–h) and 4 V (Fig. 6i–p) ependymal surfaces gradually lost all RC2 expression and had increasing GFAP expression. At birth, the GFAP + stripe corresponding to the 3 V E2 epithelium began, as only a few GFAP + cells located near the ventral midline (Fig. 6a arrowheads). This expression continued to expand dorsally until P10, when the GFAP + stripe appeared complete (Fig. 6d

**Figure 3 | Combination of molecular markers distinguish E cell types in the 3 and 4 V.** (a–e) Differential molecular marker expression in the E1, E2 and E3 epithelia in the ventral 3 V. Dashed box on the 3 V map depicts the region shown. Vimentin (**a**) and Nestin (**b**) are expressed throughout all 3 epithelia, whereas S100β (**c**) and CD24 (**d**) are expressed only in the E1 and E2 epithelia. GFAP (**e**) is expressed along a distinctive band, from anterior–ventral to posterior–dorsal, corresponding to the E2 epithelium with expression also in a few E1 cells, but no E3 cells. Roman numerals (II) and (III) indicate the position of high-power images of the E2 and E3 epithelia, respectively, shown below. Compass indicates posterior (P) and ventral (V). Scale bar, 0.25 mm. (a.II/a.III–e.II/e.III) Apical surface confocal images of E2 (a.II–e.II) and E3 (a.III–e.III) cells confirm that Vimentin (a.II,a.III) and Nestin (b.II,b.III) are expressed by E1, E2 and E3 cells; S100β (c.II,c.III) and CD24 (d.II,d.III) are expressed by E2 and E1 cells; GFAP (e.II,e.III) is expressed primarily by E2 cells, but also by a few E1 cells, but no E3 cells. γ-Tubulin or α-tubulin (red) staining identifies the E-cell type. Scale bar, 10 μm. (f–j) Low-power images of the midline 4 V floor corresponding to the dashed box on the 4 V map. Dotted lines delineate the E1–E2 boundary bilaterally. Vimentin (**f**), Nestin (**g**), S100β (**h**) and CD24 (**i**) are expressed in the E1 and E2 epithelia, whereas GFAP (**j**) is expressed primarily by E2 cells along the ventral midline. Although a small subset of E1 cells express GFAP, the expression lateral to the dotted lines belongs primarily to subependymal astrocytes, captured in these low power images. Roman numerals (II) indicate the position of high-power images shown below. Compass indicates the midline (M) and lateral (L) directions. Scale bar, 50 μm. (f.II–j.II) Confocal images of the apical surface at the E1–E2 boundary confirm the expression patterns seen in low-power images: GFAP (j.II) is seen primarily in E2 cells with only a few E1 cells labelled. It is also noteworthy that not all E2 cells express GFAP. γ-Tubulin or α-tubulin (red) staining identifies the E-cell type. Scale bar, 10 μm. (**k,l**) Percentage of E1, E2 or E3 cells in the 3 V (**k**) or 4 V (**l**) expressing the molecular markers listed. More than 100 cells counted per cell type/molecular marker from N = 3 mice (individual mouse data shown).

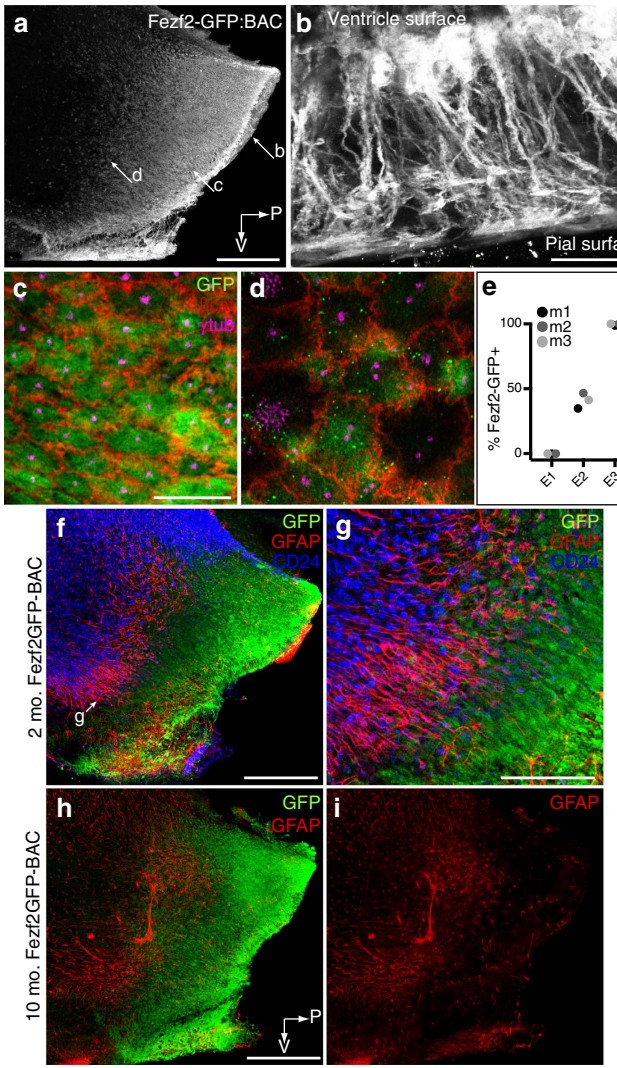

**Figure 4 | E3 cells and a subset of E2 cells express the transcription factor Fezf2.** (**a**) Whole mount (3 V) from a Fezf2-GFP:BAC transgenic mouse (GENSAT) stained with GFP antibodies indicates Fezf2 expression in the E3 and E2 territories of the 3 V infundibular recess. Compass indicates ventral (V) and posterior (P) directions. Scale bar, 0.25 mm. (**b**) At higher magnification, the ventral cut edge of the whole mount in **a** reveals the elongated morphology of Fezf2-GFP+ cells, with their apical surface and cell body facing the ventricle and their long basal process terminating at the pia. Scale bar, 50 μm. (**c,d**) Confocal images at the apical surface of the 3 V wholemount shown in **a** labelled with antibodies to GFP, β-catenin and γ-tubulin show that nearly all E3 cells (**c**) and a subset of E2 cells (**d**) express Fezf2. Scale bar, 10 μm. (**e**) Percentage of E1, E2 or E3 cells expressing Fezf2-GFP. More than 100 cells analysed per E-cell type from N = 3 mice with data shown for individual mice m1–m3. (**f,g**) Fezf2-GFP:BAC 3 V whole mounts labelled with antibodies to GFP, GFAP and CD24 show distinct molecular marker expression patterns allowing identification of the three epithelial cell types: E1 are CD24+GFAP±Fezf2−; E2 are CD24+GFAP+Fezf2±; E3 are CD24-GFAP-Fezf2+. Scale bar, 0.25 mm (**f**), 100 μm (**g**). (**h,i**) GFP and GFAP labelling of a 3 V whole mount from a 10-month-old Fezf2-GFP:BAC mouse revealed that the 3 V E2 and E3 territories are relatively stable with age. Scale bar, 0.25 mm (**h,i**).

arrowheads with dashed arc). Similarly, in the 4 V, GFAP expression increased over the first 10 days to label the ventral midline. Interestingly, 3 V GFAP expression was observed in the putative E2 region as early as P0 and P2, well before the first cells

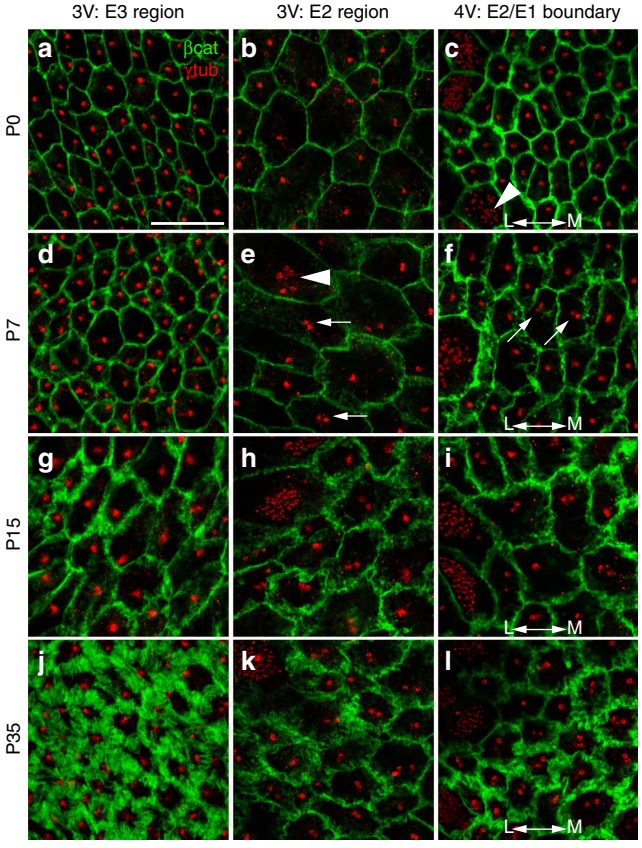

**Figure 5 | Progressive postnatal maturation of E2 and E3 apical specializations.** Apical surface confocal images in the E3 (first column) and E2 (second column) regions of 3 V whole mounts and at the E2–E1 boundary (third column) of 4 V whole mounts from postnatal ages P0 (**a–c**), P7 (**d–f**), P15 (**g–i**) and P35 (**j–l**) (three mice studied per age). Whole mounts were labelled with γ-tubulin (red) and β-catenin (green) antibodies. At P0, apical surfaces in both E3 and E2 regions of the 3 V and E2 region of the 4 V had smooth membranes and a single basal body, although differentiating E1 cells laterally in the 4 V already displayed multiple basal bodies (arrowhead in **c**). By P7, a few differentiating E2 cells in the 3 and 4 V were presenting two basal bodies (arrows in **e,f**). Some differentiating E1 cells displayed deuterosome structures (arrowhead in **e**). By P15, E3 and E2 cells had increasing membrane interdigitations (**g–i**) and most E2 cells in the 3 and 4 V were presenting two basal bodies. By P35, E3 and E2 cells' apical specializations had matured to their adult form, with more interdigitations among all cells and two basal bodies at the surface of almost all E2 cells. Lateral (L) and Medial (M) directions indicated at the 4 V surface. Scale bar, 10 μm.

with two basal bodies were seen at P7. At five ages from P0 to P10, we quantified the proportion of GFAP+ cells with either 1 or 2 basal bodies in three high-power fields covering the ventral-most aspect of the 3 V GFAP+ stripe. Greater than 95% of GFAP+ ventricle-contacting cells had a single basal body through P4. By P7, this number had declined with a corresponding increase in GFAP+ cells with two basal bodies (Fig. 6q), consistent with data on apical surface maturation. This suggests that E2 differentiation occurred over a protracted period, beginning with RC2 downregulation and GFAP upregulation in the first postnatal week, but not complete until apical specializations had matured over the first month. These observations are consistent with RC2+ radial glia progressively giving rise to E2 and E3 cells.

Consistent with the anterior to posterior gradient of RC2 downregulation and GFAP upregulation in the postnatal ventral

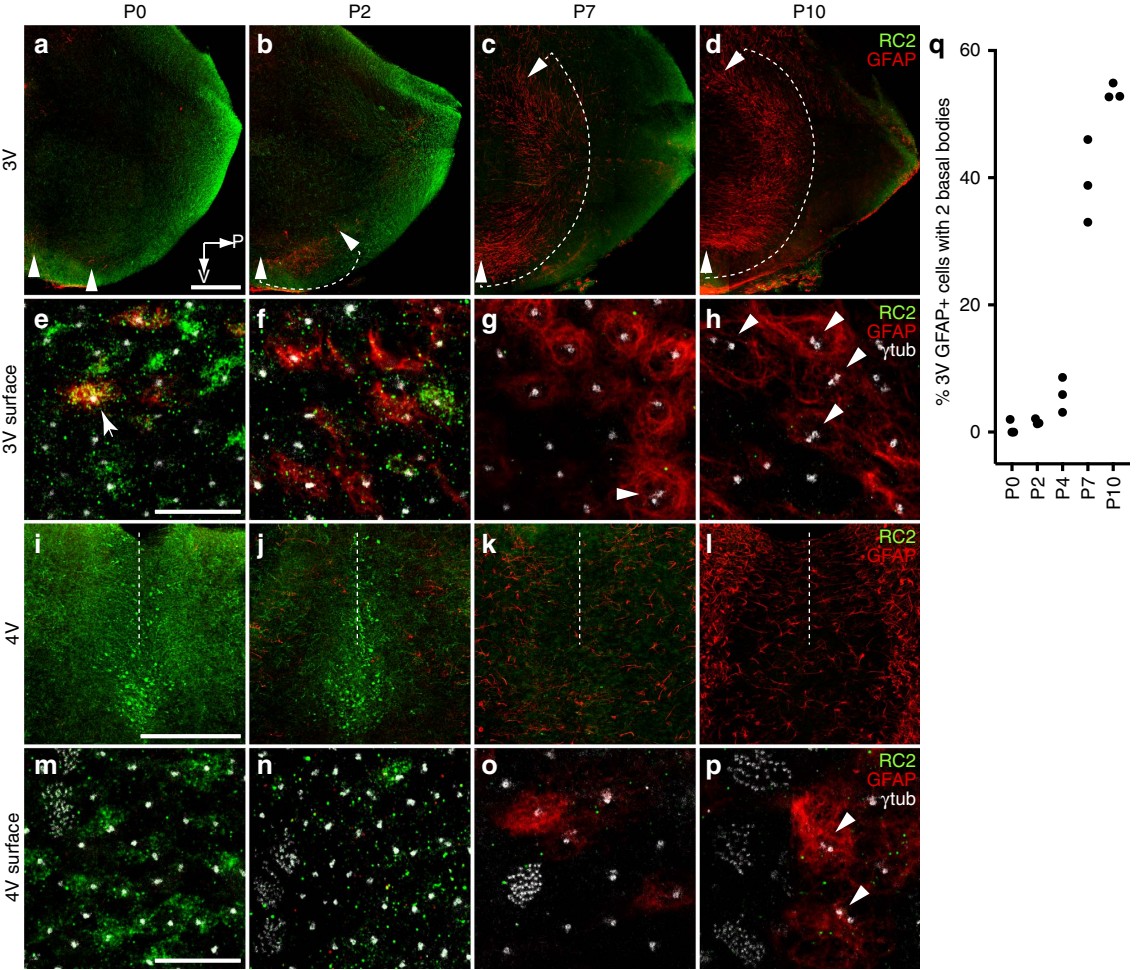

**Figure 6 | Progressive postnatal transformation of radial glia into E2 and E3 cells.** (**a**-**d**) Low-magnification wholemount views of the 3 V infundibular recess showing RC2 (green) and GFAP (red) expression at P0 (**a**), P2 (**b**), P7 (**c**) and P10 (**d**). Over this period, RC2 is downregulated in the E1, E2 and E3 epithelia, whereas GFAP is upregulated from ventral to dorsal within the E2 epithelium (arrowheads with dashed arc show expanding domain). (**e**-**h**) High-magnification images at the ependymal surface in the ventral E2 band, corresponding to wholemounts in **a**-**d**, respectively, showing γ-tubulin (white) expression. Arrow in **e** shows a cell with a single basal body expressing both RC2 and GFAP, likely to be in transition. GFAP expression is seen in multiple cells at P2 (**f**) and P7 (**g**) that have single basal bodies, with a few GFAP+ cells at P7 showing two basal bodies (arrowhead). By P10, an increasing number of GFAP+ cells display two basal bodies (arrowheads). This suggests that during E2 maturation, GFAP expression precedes apical display of the two basal bodies. Scale bars, 0.25 mm (**a-d**) and 10 μm (**e-h**). (**i-l**) Low-magnification whole-mount views of the midline 4 V floor showing similar postnatal time course of RC2 downregulation and GFAP upregulation in the 4 V. Dashed lines indicate the midline. At P10 (**l**), GFAP expression along the midline appears less robust than more laterally. However, most GFAP+ cells located laterally in these images were subependymal astrocytes (incorporated into the low power z-stack projection), whereas the midline GFAP+ cells corresponded to E2 cells. This was confirmed in high-power images at the apical surface showing γ-tubulin (white, **m-p**). As in the 3 V, GFAP+ E2 cells at P7 usually had a single basal body whereas by P10, many had two basal bodies (arrowheads in **p**). Sale bars, 0.25 mm (**i-l**) and 10 μm (**m-p**). (**q**) Percentage of GFAP+ cells displaying two basal bodies in the 3 V E2 epithelium during early postnatal development. Greater than 100 GFAP+ cells analysed from N = 3 mice per age with data shown for individual mice. GFAP+ E2 cells have a single basal body until around 1–2 weeks postnatally when their apical surfaces become progressively more specialized and display two basal bodies.

3 V, we also observed a decline in proliferation (Supplementary Fig. 6). Previous studies have indicated that proliferating cells persist postnatally in the walls of the 3 V (refs 15,32–34). It has been suggested that some of these ventricular epithelial cells retain neural stem cell properties and can generate new neurons postnatally[15,34]. However, we found very little proliferation in the ventral 3 V in adult whole mounts. We used antibodies to Ki67, a marker of cycling cells, to determine whether ependymal cells in the different 3 V territories continued to divide in the postnatal mouse brain (Supplementary Fig. 6a). At P0, a time when very few cells had acquired their specialized apical features, we found ~2,000 proliferating cells (bilaterally) in the ventral 3 V walls (Supplementary Fig. 6b). The majority of these cells were in

the caudal ventral infundibular region in a territory that will, in adults, be populated by E3 cells. Both brightly labelled (mitotic) and more lightly labelled (interphase) Ki67+ cells were found in contact with the ventricle in this region (Supplementary Fig. 6c). However, the number of proliferating cells decreased rapidly and by P28, very few Ki67+ cells were present. The few labelled cells at P28 appeared to be subependymal and in the ventral infundibular recess within the E3 territory.

**E2/E3 cells in the 3 and 4 V are floor-plate derivatives.** Finding a distinct, continuous epithelium of E3 and E2 cells along the ventral midline, extending caudally from the diencephalon,

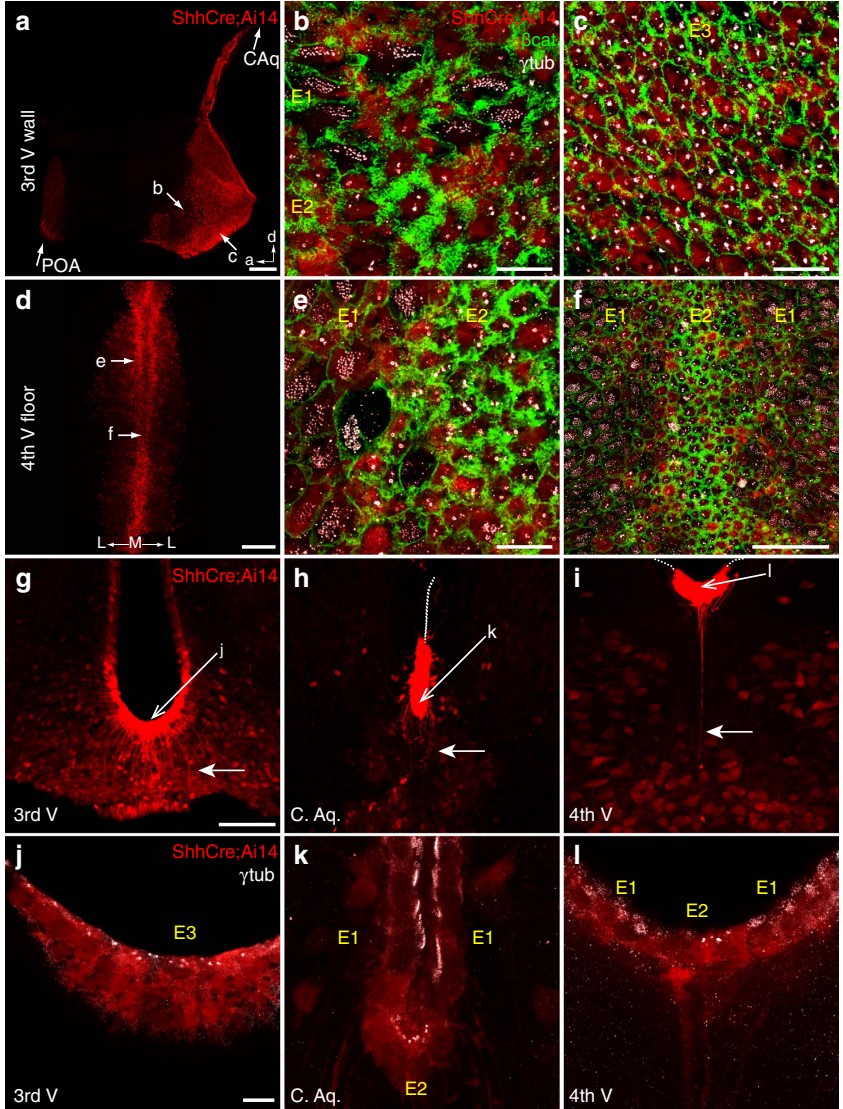

**Figure 7 | E2 and E3 territories in the ventral midline are derived from the Shh-expressing floor plate.** In low- (**a**) and high-magnification (**b**,**c**) 3 V whole-mount views, E2 (**b**) and E3 (**c**) cells are labelled in ShhCre;Ai14 mice, which express tdTomato (red) in cells expressing Cre under the Shh promoter, active in the embryonic floor plate. γ-Tubulin (white) and β-catenin (green) show basal bodies and membrane interdigitations used to identify E-cell types. A small subset of E1 cells (**b**) at the E1–E2 boundary are also tdTomato. Anteriorly located E2 and E3 cells near the POA are labelled, corresponding to the anterior-most extent of the floor plate, as well as the E2 epithelium back into the CAq. The labelled E2 epithelium continues into the 4 V floor in low- (**d**) and high-magnification (**e**,**f**) views, along with the subset of labeled E1 cells at the E1–E2 boundary. Coronal sections (**g**–**l**) show tdTomato + E2 and E3 cells with long basal processes (solid arrows in **g**–**i**). At higher magnification (**j**–**l**), co-staining with γ-tubulin shows the basal bodies in this ventral epithelium derived from ShhCre-expressing progenitors. Five mice studied in whole-mount preparations and three mice in sections. Scale bars, 0.25 mm (**a**,**d**), 10 μm (**b**,**c**,**j**–**l**), 50 μm (**f**,**g**–**i**).

suggested a unique developmental origin for this epithelium. The floor plate, comprised specialized radial glia located at the ventral midline of the neural tube, is a key organizing centre that profoundly influences development of the vertebrate nervous system[35]. Floor-plate cells secrete Shh and thereby govern glial and neuronal cell-fate specification of adjacent, dorsally located cells. We hypothesized that the floor plate could give rise to E3 and E2 epithelia.

We crossed mice expressing Cre from the Shh locus[36] to Cre-reporter mice expressing tdTomato[37] to lineage-trace floor-plate cells. Whole mounts and coronal sections were prepared from all adult ventricles and stained with γ-tubulin, β-catenin and dsRed antibodies. In whole mounts (Fig. 7a,d) and in sections (Fig. 7g–i), a ventral midline stripe of tdTomato + cells was

observed beginning in the POA and extending caudally through the 3 V, CAq and 4 V. TdTomato labelling in the POA corresponded to E2 and E3 cells identified in this region on our 3 V map. Just caudal to this area, above the optic chiasm, there was discontinuity in the tdTomato + stripe. Absence of labelling in this area corresponded to absence of E2/E3 cells in this region on our 3 V map. Caudally, high-power imaging of apical surfaces in both whole mounts and sections revealed that the tdTomato + stripe contained E2 and E3 cells in the 3 V (Fig. 7b,c,j), and E2 cells in the CAq (Fig. 7k) and 4 V (Fig. 7e,f,l). In addition, the domain of Shh-expressing cells extended laterally, just beyond the E2–E1 boundary, such that the most medially located E1 cells were also tdTomato + . Finally, tdTomato + E2 and E3 cells in sections had long basal processes penetrating the parenchyma

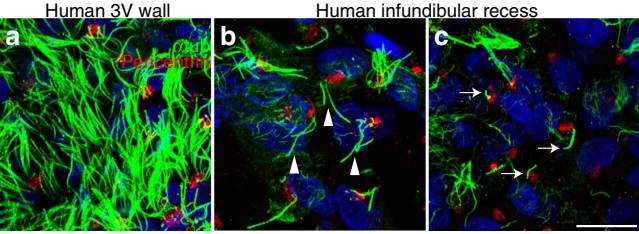

Human 3V wall Human infundibular recess

**Figure 8 | E2 and E3 cells are present in the ventral 3 V in the human brain.** Whole mounts from the 3 V walls and infundibular recess of three autopsied brains were stained with pericentrin (red), to label basal bodies, and acetylated α-tubulin (green), to label cilia. Dorsally, the 3 V wall (**a**) is covered by multiciliated ependymal cells, whereas in the infundibular recess (**b**,**c**), apical surfaces display either two long cilia (arrowheads, **b**) or single, short cilia (arrows, **c**). Scale bar, 10 μm.

(Fig. 7g–i arrows), consistent with our previous morphological analysis. This data suggested that ventral midline E2 and E3 epithelia represented the final fate of the Shh-expressing floor plate in the brain.

**Multiple E cell types in the human brain infundibular recess**. To determine whether these cell types were present in humans, we stained 3 V whole mounts, derived from autopsied human brains ($n = 3$), with antibodies for pericentrin (multiple γ-tubulin antibodies did not stain human tissue) to label basal bodies and with acetylated-tubulin to label cilia (Fig. 8). In the dorsal 3 V, multiciliated (E1) cells covered the ventricular surface. However, ventrally in the walls of the infundibular recess, we observed three cell types: multiciliated (Fig. 8a), biciliated (Fig. 8b arrowheads) and uniciliated cells (Fig. 8c arrows) that probably corresponded to E1, E2 and E3 cells, respectively. Owing to the size of the human 3 V and difficulty obtaining contiguous segments of high-quality tissue, it was difficult to characterize the epithelial pattern of E cell types, as in mice. The cell types in humans appeared less segregated than in mice. This observation suggests, however, that distinct ependymal cell types in the ventral midline, as defined by their apical specializations, are conserved between human and mouse.

## Discussion

Using high-resolution confocal microscopy of ependymal cells' apical compartment, we describe an extensive territory along the ventral 3 V, CAq and 4 V floor containing only E2 cells. In the 3 V, this E2 territory is juxtaposed to another territory of E cells (E3) displaying only a single primary cilium. We provide marker expression patterns that define these territories of specialized E cells. The work indicates that these cells express the transcription factor FezF2 and are derived from uniciliated RC2 + radial glia within Shh-expressing domains that map to the anterior floor plate. The precise organization of these unique cell types in the ventral midline suggests that early neuroepithelial patterning is reflected in the adult ependyma and probably plays a key role in the diverse and coordinated functions of the mature epithelium. More broadly, this work demonstrates a method for cataloging epithelial cell types based on their apical specializations.

The distribution of E2 and E3 apical specializations in the 3 V suggests these cells correspond to subtypes of tanycytes[21,23,24]. E2 and E3 cells have extensive membrane interdigitations[20] and long radially directed processes that penetrate the underlying neuropil[1](Supplementary Fig. 7). However, ultrastructural studies have not reported the unusual basal body found in E2 cells and scanning electron microscope studies typically report no cilia on tanycytes[20,38]. We found that almost all E3 cells have a primary cilium and most E2 cells have one or two long cilia. No region in the 3 V was uniformly non-ciliated. The difference between our results and previous findings may be due to the detection method. Using scanning electron microscope, tanycyte cilia may be difficult to distinguish amidst apical cytoplasmic protrusions and abundant microvilli. Tanycytes are classified into four subtypes (α1, α2, β1 and β2), based on location along the 3 V dorsoventral axis and basal process projection into the hypothalamus[12]. They are also distinguished by molecular markers, including GFAP, reported in α2 tanycytes[39]. Our data suggest 3 V E2 cells correspond to α-tanycytes, while E3 cells correspond to β-tanycytes (Supplementary Fig. 7). However, this pairing introduces inconsistencies in traditional tanycyte classification. First, the strictly dorsal-ventral distinction of α- and β-subtypes in coronal sections implies that both subtypes are present throughout the anterior–posterior extent of the 3 V. The E3 epithelium (β-tanycyte), however, occupies a progressively more dorsal position towards the caudal 3 V infundibular recess, where there is diminution of the E2 (α-tanycyte) domain. Second, GFAP is reported as an α2 tanycyte marker[39] with α1 cells dorsal to the GFAP+ domain, but we find only multiciliated (E1) cells dorsal to this domain. The characteristics we describe for E2 and E3 cells robustly identify these subpopulations, which will be key to understanding their distinct functions. This is demonstrated by recent work on the elusive identity of hypothalamic neural progenitor cells, which different groups suggest correspond to either the α- (ref. 39) or β-(ref. 15) tanycyte subtype. The key apical features that distinguish E2 and E3 cells we describe here may help resolve this dilemma. We see that proliferation in the ventral 3 V decreases drastically within the first month of life and this decrease seems to be regionally linked to the conversion of radial glia into different types of E cells. Notably, the areas where we observed higher number of Ki67 + cells are in territories associated with E3 cells. These cells retain a simpler apical organization, with primary cilia similar to that present in radial glia and V-SVZ B1 cells. It is therefore tempting to speculate that E3 cells (β-tanycytes) could retain a proliferative potential, despite the observed postnatal decline in proliferation. It is noteworthy that most studies of neural progenitor proliferation in the hypothalamus show dividing cells in response to an exogenous stimulus, including FGF2 (ref. 39), CNTF[32], IGF1 (ref. 40) or high-fat diet[15]. We cannot exclude, therefore, that exogenous stimulation with growth factors may induce some of these cells to proliferate; it is possible these cells have latent neural progenitor capacity.

We provide evidence for the origin of E2 and E3 cells. The floor plate is a well-known organizing centre in neural development[35], but the fate of floor-plate cells themselves is unknown. Using Shh-Cre lineage tracing, we showed that floor-plate cells transform into a unique ventral midline epithelium in the adult brain ventricles. Medially, this epithelium contained E2 and E3 cells in the 3 V and E2 cells in the CAq and 4 V; laterally, there was a subset of labelled E1 cells. During development, there is a distinction between medial and lateral floor plate: medial cells express Shh, axon guidance molecule netrin-1 and the transcription factor forkhead box A2, and lateral cells express a subset of these factors and other markers[35]. This medial–lateral floor-plate distinction may be reflected in different adult E cell types (E2/E3 (medial) versus E1 (lateral)). However, a prior study showed that neural tube Shh expression might extend ventrolaterally beyond the floor plate[41]. We therefore cannot exclude the possibility that laterally located, labelled E cells are derived from Shh-expressing cells outside the floor-plate boundary. Heterogeneity also exists in the anterior–posterior dimension of the floor plate: anterior floor-plate cells (POA to

rostral hindbrain) are distinct from posterior floor-plate cells (caudal hindbrain through spinal cord)[35]. This anterior–posterior variation may be reflected in the adult ependyma: the ventral midline stripe of E2 cells extends from the 3 V to the rostral 4 V, but caudally (Fig. 1h, caudal to asterisk), E2 cells are not limited to the ventral midline. The spinal cord E2 epithelium encircles the entire circumference of the central canal[25], consistent with a ventral origin for this epithelium[42]. E3 cells, found only in the hypothalamic region, may correspond with unique diencephalic floor-plate cells expressing bone morphogenetic protein 7 (ref. 43). In sum, medial–lateral and anterior–posterior heterogeneity in the floor plate may be inherited by the adult brain E2/E3 epithelia.

E2 cells have $(9 + 2)$ cilia that are presumably motile[25]. However, with at most two cilia and a segregated distribution along the ventral midline, E2 cells are unlikely to contribute significantly to CSF flow. E2 cells have large basal bodies with extensive appendages and their cilia are engulfed in an apical membrane invagination. A similar invagination, called the ciliary pocket, is present at the base of primary cilia[44]. Primary cilia are cellular antennae, essential to the transduction of diverse extracellular signals, such as light, and intercellular signals, such as Shh[45]. The basal body and transition zone of cilia are critical for signal transmission to and from the cell body[46–49]. Both E3 cells, which have primary cilia, and E2 cells, which have one to two motile cilia with features of primary cilia, probably have sensory functions. The elaborate E2 basal body may be involved in transmitting information between the cilium and intracellular compartments. The long process of both cell types may help transmit these signals to underlying neural circuits. The E2 territory, spanning from the 3 V through the CAq to the 4 V, provides a continuous conduit for information transfer from the ventricle to the underlying brain between E2 cells' cilia and their basal processes.

E2 and E3 cell functions may be regionally specified. In the 3 V, the correspondence of these cells with tanycytes suggests they may function as postnatal neural progenitor cells[15,16,34] for hypothalamic neurons controlling energy balance. Tanycytes have been described as gatekeepers of the mediobasal hypothalamus[13,14], regulating access of metabolic signals to this area. E2 and E3 cilia may participate in both functions: regulating progenitor activity and sensing CSF metabolites. Identifying these apical specializations in the human brain suggests that further exploration of their function could yield insights into central control of human metabolism. In the 4 V, E2 cells overly the raphe nuclei, where they may provide feedback to serotonergic neurons that project intraventricular axons controlling neurogenesis in the V-SVZ[50]. Interestingly, we found that E2 and E3 cell maturation occurs over a protracted postnatal period, suggesting these functions may be linked to postnatal maturation. The identification of an extensive epithelium of these cells amenable to transgenic manipulations provides a tractable system where these hypotheses may be tested.

## Methods

**Mice.** CD1 mice (Charles River) 2–4 months old (adult studies) or E14.5–P35 (developmental studies) were used to study the ventricular surface; 2- to 4-month-old Fezf2-GFP-BAC mice[26] were used to study Fezf2+ cells; Shh-Cre/GFP mice[36] were crossed with a tdTomato Cre reporter line[37] and progeny were used at 2 months old, to lineage trace floor-plate cells (both sexes used for all studies; mice on 12 h:12 h light–dark cycle in five per cage group housing). The Institutional Animal Care and Use Committee approved all animal procedures.

**Human specimens.** Three brains (ages 62, 64 and 71 years with post mortem interval 16, 9 and 12 h, respectively) were collected at autopsy. Multiple whole mount blocks were dissected from the third ventricle ependymal wall from each brain, their positions along the wall were documented and the tissue was immersion-fixed in 4% paraformaldehyde at 4 °C for 24 h. All specimens were collected with informed consent and in accordance with the St Joseph's Hospital and Medical Center Committee on Human Research (IRB number 10BN159).

**Whole mount dissection.** After cervical dislocation, the mouse brain was removed from the skull and 3 V whole mounts were freshly dissected using priniciples similar to those described for LV whole mounts[51]. Briefly, 3 V whole mounts were dissected by performing a ventriculotomy of the 3 V from ventral to dorsal and rostral to caudal; 4 V whole mounts were dissected by removing the cerebellum, caudal to rostral, to expose the 4 V floor. The exposed ventricle walls were immersion-fixed in 4% paraformaldehyde at 4 °C overnight.

**Immunostaining and microscopy.** Primary and secondary antibodies were incubated in PBS with 0.5% Triton-X100 (surface antigens) or 2% Triton-X100 (deep antigens) and 10% normal goat serum for 24 h (surface antigens) or 48 h (deep antigens) at 4 °C. Primary antibodies were as follows: mouse anti-acetylated tubulin (1:1,000, Sigma T6793), rabbit anti-γ-tubulin (1:1,000, Sigma T5192), mouse anti-γ-tubulin (1:500, Abcam ab11316), mouse anti-β-catenin (1:500, BD Transduction Laboratories 610153), rabbit anti-β-catenin (1:1,000, Sigma C2206), rat anti-CD24 (1:500, BD Pharmingen 557436), rabbit anti-S100β (1:500, Dako Z0311), chicken anti-vimentin (1:500, EMD Millipore AB5733), chicken anti-Nestin (1:200, Lifespan Biosciences, LS-B5224), mouse anti-GFAP (1:500, EMD Millipore MAB3402), rabbit anti-dsRed (1:1,000, Clontech 632496), rabbit anti-pericentrin (1:500, Abcam ab4448), mouse anti-RC2 (1:100, Developmental Studies Hybridoma Bank, 0.1 ml concentrate) and chicken anti-GFP (1:500, Aves Labs GFP-1020). Secondary antibodies were as follows: conjugated to Alexa Fluor dyes (goat or donkey polyclonal, 1:400, Invitrogen). Confocal images were taken on a Leica SP5 or SPE. Transmission electron microscopy analysis was performed as described previously[52]. For reconstruction of the apical surface and basal bodies of E cells, we cut ~600 serial ultrathin (0.05 µm) sections that were placed on Formvar-coated single-slot grids, stained with lead citrate and examined under a Jeol 100CX EM.

**Ependymal surface maps.** Tiled high-resolution confocal images ($116.4 \times 116.4 \mu m^2$) were taken using a $\times 63$ oil objective (numerical aperture 1.47), to reconstruct the entire ependymal surface from 13 V wall (935 tiled images) and 14 V floor (793 tiled images). These reconstructions were then imported into Illustrator CS5 (Adobe), where the wall was outlined and the location of E2 and E3 cells plotted.

**Quantification of basal bodies in postnatal 3 V GFAP + cells.** In three high-power fields ($116.4 \times 116.4 \mu m^2$) from the ventral-most aspect of the E2 GFAP + band in the 3 V, we counted the number of basal bodies per GFAP + apical surface. We chose to quantify this ventral-most region, because that is where GFAP expression initiated at P0. These counts were performed in three mice at P0, P2, P4, P7 and P10.

**Quantification of apical surface specializations.** Cilia complement and length, apical surface area and marker expression were each studied in three images ($116.4 \times 116.4 \mu m^2$) evenly distributed along the dorsoventral axis of the E2 or E3 epithelium in the 3 V or anterior–posterior axis of the E2 epithelium in the 4 V. Cilia length and apical surface area were quantified using Metamorph (Molecular Devices).

**Statistical analysis.** Use of validated antibodies allowed reduction in sample size as listed for each experiment. Descriptive statistics and $t$-tests were computed in R.

**Data availability.** Source data for 3 and 4 V ependymal surface maps, as well as all other data for figures contained in this paper, are available from the authors.

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

## Acknowledgements

Z.M. supported by NIH grant F32 DK108596 and the Barrow Neurological Foundation grant number 559035022. Additional support from NIH grants R01 NS082745 (to N.S.) and R37 HD032116 (to A.A.-B.), and a generous gift from the John G. Bowes Research Fund. A.A.-B. is the Heather and Melanie Muss Endowed Chair of Neurological Surgery at UCSF. We thank R. Fiorelli and S. Borwege for help with human specimen collection.

## Author contributions

Z.M., B.C., J.M.G.-V., N.S. and A.A.-B. designed, funded and supervised the research. Z.M., Y.K., M.D.-M., E.C., S.G.-P. and C.O. performed the research. Z.M., Y.K., M.D.-M. and S.G.-P. analysed the data. Z.M. and A.A.-B. wrote the manuscript.

## Additional information

**Competing financial interests:** The authors declare no competing financial interests.

**Publisher's note**: 

