## [Peer Review File · Nature Communications]

REVIEWERS' COMMENTS:

Reviewer #1 (Remarks to the Author):

Mirzadeh and collaborators report on their detailed study of ependymal cells in V3 and V4. They describe biciliated E2 and monociliated E3 cells, with detailed distribution, some molecular markers and data on their lineage history. Although the paper does not directly address the function of those cells, careful descriptive studies like this one are necessary.

I reviewed that manuscript previously and authors provide novel supplementary data by which they at least partly address most of my remarks. The present version is very significantly improved and stands as a solid reference and basis for future physiological studies. I believe that this paper is of sufficient general interest for Nat Communication.

POINT-BY-POINT RESPONSE TO REVIEWERS:

REVIEWERS' COMMENTS:

Reviewer #1 (Remarks to the Author):

Mirzadeh and collaborators report on their detailed study of ependymal cells in V3 and V4. They describe biciliated E2 and monociliated E3 cells, with detailed distribution, some molecular markers and data on their lineage history. Although the paper does not directly address the function of those cells, careful descriptive studies like this one are necessary.

I reviewed that manuscript previously and authors provide novel supplementary data by which they at least partly address most of my remarks. The present version is very significantly improved and stands as a solid reference and basis for future physiological studies. I believe that this paper is of sufficient general interest for Nat Communication.

We were very encouraged by this reviewer's positive comments and we believe the changes we made previously to address his concerns have strengthened the work.